# c-MYC Protein Stability Is Sustained by MAPKs in Colorectal Cancer

**DOI:** 10.3390/cancers14194840

**Published:** 2022-10-04

**Authors:** Martina Lepore Signorile, Valentina Grossi, Candida Fasano, Giovanna Forte, Vittoria Disciglio, Paola Sanese, Katia De Marco, Francesca La Rocca, Raffaele Armentano, Anna Maria Valentini, Gianluigi Giannelli, Cristiano Simone

**Affiliations:** 1Medical Genetics, National Institute of Gastroenterology Saverio de Bellis, IRCCS Research Hospital, Castellana Grotte, 70013 Bari, Italy; 2Department of Pathology, National Institute of Gastroenterology Saverio de Bellis, IRCCS Research Hospital, Castellana Grotte, 70013 Bari, Italy; 3Scientific Direction, National Institute of Gastroenterology Saverio de Bellis, IRCCS Research Hospital, Castellana Grotte, 70013 Bari, Italy; 4Medical Genetics, Department of Biomedical Sciences and Human Oncology (DIMO), University of Bari Aldo Moro, 70124 Bari, Italy

**Keywords:** p38α, MAPKs, c-MYC, protein stabilization, kinase activity, CRC

## Abstract

**Simple Summary:**

Colorectal cancer (CRC) is the most common gastrointestinal tract malignancy. Previous reports have shown that cancerous phenotypes in the intestine are dependent on c-MYC target gene expression. Unfortunately, finding c-MYC inhibitors has proven difficult because c-MYC does not have a deep surface-binding pocket. Considering that c-MYC is maintained upregulated through β-catenin-mediated transcriptional activation and ERK-mediated post-translational stabilization, and since we have previously demonstrated that c-MYC transcriptional activation is affected by p38α as a β-catenin chromatin-associated kinase, here, we investigated p38α’s involvement in c-MYC protein stabilization in CRC. Interestingly, we found that p38α sustains c-MYC’s stability by preventing its ubiquitination and proteasomal degradation. Moreover, we showed that p38α inhibitors exhibit a synthetic lethality effect when used in combination with MEK inhibitors in CRC cells. Our findings identify p38α as a promising therapeutic target that acts on the pharmacologically “undruggable” c-MYC protein, with implications for countering c-MYC-mediated CRC proliferation, metastasization, and chemoresistance.

**Abstract:**

c-MYC is one of the most important factors involved in colorectal cancer (CRC) initiation and progression; indeed, it is found to be upregulated in up to 80% of sporadic cases. During colorectal carcinogenesis, c-MYC is maintained upregulated through β-catenin-mediated transcriptional activation and ERK-mediated post-translational stabilization. Our data demonstrate that p38α, a kinase involved in CRC metabolism and survival, contributes to c-Myc protein stability. Moreover, we show that p38α, like ERK, stabilizes c-MYC protein levels by preventing its ubiquitination. Of note, we found that p38α phosphorylates c-MYC and interacts with it both in vitro and in cellulo. Extensive molecular analyses in the cellular and in vivo models revealed that the p38α kinase inhibitors, SB202190 and ralimetinib, affect c-MYC protein levels. Ralimetinib also exhibited a synthetic lethality effect when used in combination with the MEK1 inhibitor trametinib. Overall, our findings identify p38α as a promising therapeutic target, acting directly on c-MYC, with potential implications for countering c-MYC-mediated CRC proliferation, metastatic dissemination, and chemoresistance.

## 1. Introduction

Colorectal cancer (CRC) is the most frequent malignant tumor affecting the gastrointestinal tract and the third most common cancer in both men and women [1,2]. Worldwide, about 2 million people each year will develop CRC (WORLD HEALTH ORGANIZATION -2022- https://www.who.int/news-room/fact-sheets/detail/cancer; accessed on 10 June 2022), with a substantial variation in five-year relative survival rates, depending on the disease’s stage at diagnosis. Eighty-five percent of sporadic CRCs progress through the classical adenoma–carcinoma sequence, showing a lack of allelic balance at several chromosomal loci, such as 5q (APC), 18q (DCC/SMAD4), and 17q (p53), and chromosomal amplification and translocation, which together contribute to tumor aneuploidy and chromosomal instability [3,4,5]. Activation of the Wnt pathway has been linked to CRC as a result of the identification of abnormalities in chromosome 5q, which were recognized as early events in the carcinogenic process for sporadic and hereditary tumors. The 5q region harbors the *APC* gene, whose protein product is a key component of the β-catenin destruction complex, together with GSK3β, axin, and CK1. During CRC carcinogenesis, stabilized β-catenin exerts its oncogenic role by activating the transcription of several regulatory genes, including *c-MYC* [6,7,8]. Surprisingly, in 2007, Sansom et al. found that, after APC loss, nuclear β-catenin was insufficient to induce any phenotype in the absence of c-MYC. Indeed, over half of the Wnt targets that showed significant induction after APC loss were no longer upregulated in the absence of c-MYC [9,10]. These data indicate that the phenotypes acquired after APC loss in the intestine greatly depend on c-MYC target gene expression [9,10]. c-MYC plays a major role in the onset and progression of CRC, where it is overexpressed in up to 80% of sporadic cases [10]. Two different mechanisms are involved in c-MYC upregulation in CRC: transcriptional activation by β-catenin due to APC inactivation, and post-translational stabilization due to serine 62 (S62) phosphorylation, mediated by ERK, a KRAS downstream kinase [9,11]. Indeed, the stability of the c-MYC protein is controlled by phosphorylation at two specific sites: S62 and threonine 58 (T58). Phosphorylation at T58 but not at S62 makes c-MYC prone to subsequent proteasome degradation [12]. The importance of ERK activation in intestinal tumorigenesis was highlighted by Lee and colleagues by using the APC^Min/+^ murine model [13]. These mice developed multiple intestinal neoplasias (Min) after spontaneously losing the heterozygous wild-type *APC* allele in intestinal epithelial cells and consequently died by the time they reached 6 months of age [13,14]. The authors revealed that the activation of ERK in these cells is essential for tumor growth via the stabilization of c-MYC protein [13]. The MEK/ERK signaling pathway has been frequently found to be overactive in CRC due to activating mutations in the upstream kinases, RAF (15%) and RAS (50%) [15,16]. This pathway is activated by a variety of different stimuli, many of which promote cell proliferation and inhibit pro-apoptotic responses [17]. ERK belongs to the MAPK family, which is composed of six distinct groups in mammals: ERK1/2, ERK3/4, ERK5, ERK7/8, Jun N-terminal kinases JNK1/2/3, and p38 MAPKs [18,19]. In mammals, p38 MAPKs are encoded by four genes: *MAPK14* (p38α), *MAPK11* (p38β), *MAPK12* (p38γ), and *MAPK1*3 (p38δ) [20]. The strict regulation of the life/death signals by p38 MAPKs can result in opposite molecular functions during tumor growth. Currently, it is assumed that p38α has a tumor suppressor role in normal cells at the onset of cellular transformation, while it acts as an oncogene upon the acquisition of a malignant phenotype [11,21]. Our group recently showed that MEK inhibition with the specific inhibitor PD98059 triggers p38α phospho-activation in various CRC cell lines (HT29, HCT116, and Caco2 cells) with different genetic backgrounds. We also found that the MEK/ERK pathway is significantly activated in CRC cells treated with the p38α inhibitor SB202190, irrespective of the mutational status of the ERK upstream activators, RAS and RAF. Indeed, an increased phospho-activation of both MEK and ERK was observed in the SB202190-treated *BRAF*-mutant HT29 cells, *KRAS*-mutant HCT116 cells, and Caco2 cells, which are the wild-type for these two genes [21]. This effect was confirmed by using a structurally and functionally different pan-p38 inhibitor, BIRB796, and was also observed in vivo in HT29-xenografted nude mice and colon sections from azoxymethane-treated APC^Min/+^ mice injected with SB202190. Importantly, these findings showed the existence of a p38α/ERK crosstalk in CRC cells [21]. Recently, we also identified p38α as a new druggable member of the β-catenin chromatin-associated kinase complexes in colorectal model systems. Indeed, we found that p38α pharmacological inhibition induces the downregulation of several β-catenin target genes, including *c-MYC*, showing that p38α is involved in several key functions of CRC cells, including proliferation, migration, and chemoresistance [22]. Since dysregulated kinase signaling is implicated in colorectal carcinogenesis, protein kinases represent an important class of drug targets in these tumors [23]. In the search for more selective pharmaceutical compounds, new drug combinations, and treatment schedules are being assessed to enhance CRC regression, increase overall survival, and improve patients’ quality of life. These include trametinib, an orally bioavailable inhibitor of MEK kinases with potential antineoplastic activity that entered clinical trials for advanced or metastatic CRC (NCT03714958, NCT04303403, NCT01750918, NCT02900664, NCT02230553, NCT03317119, NCT03377361, NCT04294160, NCT02538627, NCT02703571, NCT05275374, NCT05358249, NCT04892017, https://www.clinicaltrials.gov; accessed on 11 June 2022) [24,25], and ralimetinib, a p38α inhibitor that is also under clinical investigation (NCT02860780, https://www.clinicaltrials.gov; accessed on 11 June 2022) for advanced or metastatic CRC [26,27]. Unfortunately, finding small-molecule or biologic inhibitors of c-MYC has proven difficult because c-MYC is localized within the nucleus and does not have a deep surface-binding pocket. Thus, considering that c-MYC is regulated at the transcriptional level by β-catenin and at the post-translational level by ERK, and since we have previously demonstrated that c-MYC transcriptional activation is affected by p38α as a β-catenin chromatin-associated kinase [22], here, we investigated the involvement of p38α in c-MYC protein stabilization.

## 2. Materials and Methods

### 2.1. Cell Culture and Reagents

HCT116, HT29, and Caco2 cells were purchased from ATCC. The HCT116 and HT29 cells were maintained in DMEM (Gibco, Carlsbad, CA, USA) supplemented with 10% fetal bovine serum (FBS) (Gibco, Carlsbad, CA, USA) and 1% antibiotics (Gibco, Carlsbad, CA, USA). The Caco2 cells were maintained in DMEM supplemented with 20% FBS and 1% antibiotics. Cells were routinely propagated under standard conditions in a 37 °C and 5% CO_2_ incubator. The p1 and p2 CRC-stem cells (CRC-SCs) were isolated from CRC patients and propagated as tumorspheres. Briefly, human colon tissue fragments were obtained in accordance with the ethical standards of the institutional committee on human experimentation. The tumor samples were subjected to mechanical and enzymatic dissociation using the tumor dissociation human kit (Miltenyi Biotec, Bergisch Gladbach, Germany) according to the manufacturer’s instructions. For magnetic separation, cells were labeled 24–48 h after dissociation with CD44 microbeads using the Miltenyi Biotec CD44 cell isolation kit. CRC-SC were maintained in DMEM/F12 Advanced (Gibco, Carlsbad, CA, USA) supplemented with 6 mg/mL Glucose (Sigma-Aldrich, St. Louis, MO, USA), 2 mM L-Glutamine (Gibco, Carlsbad, CA, USA), 10 ng/mL bFGF (Sigma-Aldrich, St. Louis, MO, USA), 20 ng/mL EGF (Sigma-Aldrich, St. Louis, MO, USA), 50× B27 supplement (Gibco, Carlsbad, CA, USA), and 100× N-2 supplement (Gibco, Carlsbad, CA, USA). All cell lines were tested to be mycoplasma-free (Minerva Biolabs, Berlin, Germany). SB202190, PD98059, PD0325901, 3-methyladenine, trypan blue, and bafilomycin were from Sigma-Aldrich (St. Louis, MO, USA). Ralimetinib (LY22228820) was kindly provided by Eli Lilly and Company (Indianapolis, IN, USA). Trametinib, cycloheximide, MG132, TWS119, and PiB were from Calbiochem (Darmstadt, Germany).

### 2.2. Quantitative Real-Time PCR 

RNA extraction and real-time PCR were performed as previously described [21]. Primer sequences are available on request.

### 2.3. RNA Interference

For RNA interference, cells were transfected with 100 nM Select Silencer siRNAs against p38α and MEK (both from Ambion by Life Technologies, Carlsbad, CA, USA) using the Hiperfect Transfection reagent (Qiagen, Hilden, Germany). On-Target plus control siRNAs (Thermo Fisher Scientific by Life Technologies, Carlsbad, CA, USA) were used as control sequences. The siRNA sequences are available on request.

### 2.4. Proliferation Assays

Proliferation assays, measuring cell metabolic activity, were performed using the WST-1 reagent (Roche, Mannheim Germany) according to the manufacturer’s instructions. Briefly, the cells were seeded in 96-well plates at a density of 5000 cells/well in 100 µL of DMEM containing 10% FBS one day before treatment. After 12 h, 24 h, 36 h, or 48 h of exposure to ralimetinib (10 μM), SB202190 (10 μM), trametinib (1 nM), or DMSO (vehicle), 10 μL of the Cell Proliferation Reagent, WST-1, were added to each well and incubated at 37 °C in a humidified incubator for 1 h. The absorbance was measured on a SPECTROstar Omega microplate reader (BMG Labtech, Offenburg, Germany) at 450–655 nm. Each assay was performed in triplicate, and the experiment was repeated three times. The proliferation index was calculated as the ratio of the WST-1 absorbance of the treated cells at the indicated time points to the WST-1 absorbance of the same experimental group at 0 h.

### 2.5. Immunoblot Analysis

The cells were collected and homogenized in lysis buffer (50 mM Tris-HCl pH 7.4, 5 mM EDTA, 250 mM NaCl, and 0.1% Triton x-100) supplemented with protease and phosphatase inhibitors and quantified using the Pierce BCA Protein Assay Kit (Thermo Fisher Scientific by Life Technologies, Carlsbad, CA, USA). An amount of 20–40 μg of protein extracts from each sample were denatured in 5× Laemmli sample buffer and subjected to 7.5% precast TGX Stain-Free polyacrylamide gel (Bio-Rad Laboratories, Munchen, Germany) electrophoresis. Proteins were electrotransferred onto nitrocellulose membrane and blocked with 1× Phosphate Buffered Saline (PBS) with 1% casein (Bio-Rad Laboratories, Munchen, Germany) for 40 min at room temperature. The immunoblots were performed using anti-β-actin, anti-c-MYC, anti-p38α MAPK, anti-MEK, anti-phospho-MAPKAPK-2 (Thr334) (p-MK2), anti-Ubiquitin, anti- β-catenin, anti-cyclin D1, anti-LC3, anti-cleaved PARP, and anti-phospho-p44/42 MAPK (Thr202/Tyr204) (all from Cell Signaling, Danvers, MA, USA). After incubation with the HRPO-conjugated secondary antibodies (GE Healthcare, Milwaukee, WI, USA), the signal was revealed using the ECL-plus chemiluminescence reagent (GE Healthcare, Milwaukee, WI, USA) according to the manufacturer’s instructions. Densitometric evaluations were carried out using the ImageJ and ImageLab software.

### 2.6. Co-Immunoprecipitation 

Co-immunoprecipitation was carried out as previously described [21]. The primary antibodies were anti-p38α and anti-c-MYC (both from Cell Signaling, Danvers, MA, USA). IgG was used as a negative control.

### 2.7. In Vitro Pull-Down Assay 

GST-p38α and HIS-β-catenin fusion proteins were generated as previously described [21]. The GST-p38α human recombinant protein was incubated with c-MYC human recombinant protein (Abcam, Cambridge, MA, USA). The HIS-β-catenin fusion protein was used as a positive control [21]. Proteins were incubated for 1 h at 4 °C on a rocking platform for in vitro binding. The fusion proteins were precipitated with Pierce Glutathione Magnetic Agarose Beads (Thermo Fisher Scientific by Life Technologies, Carlsbad, CA, USA) according to the manufacturer’s instructions and then washed extensively in buffer A (20 mM Tris-HCl pH 8, 150 mM KCl, 5 mM MgCl2, 0.2 mM EDTA, 10% glycerol, 0.1% NP-40) containing fresh inhibitors and 1 mM DTT. Afterward, the precipitates were resolved on 10% SDS-PAGE and analyzed by immunoblot. The primary antibodies were anti-polyHistidine (Sigma-Aldrich, St. Louis, MO, USA), anti-GST (Cell Signaling, Danvers, MA, USA), and anti-c-MYC (Cell Signaling, Danvers, MA, USA). Rabbit IgG HRP and Mouse IgG HRP (GE Healthcare, Milwaukee, WI, USA) were used as secondary antibodies and revealed using the ECL-plus chemiluminescence reagent (GE Healthcare, Milwaukee, WI, USA).

### 2.8. Immunohistochemistry

Tissue specimens were formalin-fixed in 4% buffered formalin, embedded in paraffin, and sectioned at 4 µm thicknesses. The sections were de-waxed and rehydrated in dH_2_O. Endogenous peroxidase activity was blocked by incubation in 3% hydrogen peroxide for 10 min. Then, the sections were mounted on Apex Bond IHC Slides (Leica Biosystems, Nussloch, Germany) and used for immunohistochemical analysis. Immunohistochemical staining procedures were carried out on a BOND III automated immunostainer (Leica Biosystems, Nussloch, Germany), from deparaffinization to counterstaining with hematoxylin, using the Bond Polymer Refine Detection Kit (Leica Biosystems, Nussloch, Germany). For c-MYC detection, a rabbit monoclonal antibody (clone D84C12; Cell Signaling, Danvers, MA, USA) was diluted at 1:100 and incubated for 30 min at room temperature. Antigen retrieval was performed using the BOND Epitope Retrieval Solution 2, a ready-to-use EDTA-based pH 9 reagent (Leica Biosystems, Nussloch, Germany). Images were acquired using a Nikon Eclipse Ti2 microscope. Ten fields with an equal area were selected for analysis at 40× magnification. Protein expression was assessed with the ImageJ software and reported as a positivity percentage.

### 2.9. In Vitro Kinase Assay 

An analysis of p38α kinase activity was performed using the ADP-Glo Kinase Assay (Promega, Madison, WI, USA) according to the manufacturer’s instructions. The p38α active protein (10 ng, Promega, Madison, WI, USA) was assayed in a kinase reaction buffer with 500 ng c-MYC human recombinant protein (Abcam, Cambridge, MA, USA), 150 µM ATP, and varying concentrations (0.01, 0.1, and 1 µM) of ralimetinib (LY22228820). A total of 500 ng of p38 peptide substrate was used as a control. The generated luminescence was measured using a SPECTROstar Omega microplate reader (BMG Labtech, Offenburg, Germany).

### 2.10. In Vivo Studies 

For the xenograft experiments, 10 × 10^6^ HT29 cells were injected subcutaneously into the flank (0.2 mL per flank, serum-free DMEM culture medium) of female athymic nude mice (Jackson Laboratory, Bar Harbor, ME, USA). The tumor volume was measured every 2–3 days using the following formula: volume (mm^3^) = (width)^2^ × length × 0.5. When the tumor volume reached 100 mm^3^, mice were randomized into four treatment groups. Mice were treated daily for 10 days with 0.05 μmol/kg of SB202190 by intraperitoneal injection (n = 8), 3 mg/kg of oral PD0325901 (n = 10), a combination of both (n = 10), or the vehicle (DMSO) alone (n = 8). After 10 days, the treatment was discontinued, animals were sacrificed, and tumors were explanted. For chemical-induced colitis-associated carcinogenesis, 20 C57BL/6 mice were injected intraperitoneally with 12 mg/kg of azoxymethane (AOM). Then, 2% dextran sulfate sodium (DSS) was given in their drinking water over five days, followed by two weeks of regular water. This cycle was repeated three times. Ten days after the last round of DSS, 10 animals were treated concomitantly with SB202190 and PD0325901 and 10 with the vehicle alone, as described above. All tissues were fixed overnight in 10% formalin and embedded in paraffin. The procedures involving animals were conducted in conformity with the institutional guidelines that comply with national and international laws and policies.

### 2.11. Cell Viability Assay

The cells were cultured in 60 mm dishes in the presence or absence of the indicated drugs. After 48 h, media were discarded, and the cells were washed twice with 1× PBS. An amount of 2 mL of Coomassie brilliant blue (Bio-Rad Laboratories, Munchen, Germany) was added to each dish for 5 min, and then the cells were washed with ethanol 70% to remove the excess Coomassie. The plates were dried at room temperature.

### 2.12. Ki67 Staining

The cells that were cultured in the presence or absence of the indicated drugs were analyzed to determine the percentage of the proliferating cells based on Ki67 expression using the Muse Ki67 Proliferation Kit (Luminex, Austin, TX, USA) according to the manufacturer’s instructions.

### 2.13. Annexin V Staining

A total of 2 × 10^4^ cells/plate, cultured in the presence or absence of the indicated drugs, were stained with the Muse Annexin V and Dead Cell Reagent (Luminex, Austin, TX, USA) according to the manufacturer’s instructions.

### 2.14. In Silico Prediction Analysis 

In silico phosphorylation prediction analysis was performed using the iPTMnet (https://research.bioinformatics.udel.edu/iptmnet accessed on 14 April 2022), NETPHOS 3.1 (https://services.healthtech.dtu.dk/service.php?NetPhos-3.1; threshold ≥ 3, accessed on 14 April 2022), Phosphosite Plus (https://www.phosphosite.org/homeAction, only HTP data, accessed on 14 April 2022), and KynasePhos 2.0 (http://kinasephos2.mbc.nctu.edu.tw/, accessed on 13 December 2020) servers.

### 2.15. Mass Spectrometry Analysis

The mass spectrometry analysis was performed by the Cogentech SRL service (Cogentech SRL, Institute of Molecular Oncology (IFOM)). The gel bands were subjected to reduction with DTT, alkylation with IAA, and digestion with Asp-N; the peptides were then loaded onto TiO_2_ resin for phospho-enrichment. Then, the flow-through was treated with StageTip C18 (for desalting), samples were enriched for phospho-peptides, and the desalted flow-through was further purified with SP3 and then analyzed by nLC-ESI-MS/MS on a Q Exactive HF mass spectrometer (Thermo Fisher Scientific) with a 32 min gradient. Samples were run in technical duplicate, in a positive mode with electrospray ionization. Data acquisition and processing were performed with Analyst TF (version 1.7.1, AB SCIEX). For the microflow analysis, the following parameters were used: CUR 30 psi, GAS1 30 psi, GAS2 30 psi, source temperature 200 °C, capillary voltage 5500 V. Spectra were acquired by a full-mass scan from 200–1800 *m*/*z* and an information-dependent acquisition (IDA) from 100–1800 *m*/*z* (top 10 spectra per cycle). Data were analyzed using the Proteome Discoverer, Mascot, and Scaffold setting software. The parameter settings of data processing were as follows: DataBase = Uniprot_CP_Human_2020_c-MYC (Database of Uniprot_cp_Human + sequence of c-MYC Abcam ab169901 Human with Accession Number P169901); Enzyme = Asp-N (cuts at the N-term of D and also on E); Modifications = Acetyl (Protein N-term), Carbamidomethyl (C), Oxidation (M), Phosphorylation (STY); Peptide Thresholds: 95.0% minimum; Protein Thresholds: 99.0% minimum; 2 peptides/protein minimum.

### 2.16. Dataset Analysis 

The association of the p38α and c-MYC protein levels (Reverse-Phase Protein Array, RPPA data) with CRC aggressiveness was analyzed based on the protein expression data (Z-scores) of 581 patients obtained from The Cancer Genome Atlas (TCGA) PanCancer Atlas through the cBioPortal website (https://www.cbioportal.org/; accessed on 2 May 2022 [28,29]). The patients were stratified based on the p38α and c-MYC protein expression data Z-scores into four groups with high (>median, n = −0.03) or low (≤median, n = −0.03) p38α protein expression and high (>median, n = −0.12) or low (≤median, n = −0.12) c-MYC protein expression. The progression-free survival (PFS) and disease-free survival (DSF) curves of the CRC patients with high p38α and high or low c-MYC protein expression levels were assessed according to the Kaplan–Meier method based on the cBioPortal online instructions. A log-rank test was performed to determine the significance of the differences between the DSF and PFS curves in the CRC patients with high (n = 77/232) and low (n = 154/232) c-MYC protein expression levels, with a *p*-value < 0.05 indicating a statistically significant difference. 

### 2.17. Quantification and Statistical Analysis

The statistical significance of the results was analyzed using the Student’s *t*-test, and a *p*-value < 0.05 was considered statistically significant. The results represent at least three independent experiments.

## 3. Results

### 3.1. Chemical Inhibition or Genetic Ablation of the MAPKs p38α and ERK Decreases c-MYC Protein Levels

The existence of a p38α/ERK crosstalk and the identification of the ERK-mediated c-MYC protein stabilization mechanisms prompted us to investigate the potential role of p38α in c-MYC regulation [21,30]. To this end, we analyzed the c-MYC protein levels upon treatment with classical p38α and MEK chemical inhibitors (SB202190 and PD98059, respectively) in the CRC cell lines (HT29, HCT116, and Caco2 cells) carrying different genetic backgrounds and that were representative of the chromosomal instability (CIN) and microsatellite instability (MIN) CRC phenotypes, as indicated in Figure 1a (left panel). Our results showed that p38α inhibition induces a decrease in the c-MYC protein levels regardless of the genetic background (Figure 1a, right panel, Appendix A). In the HT29 CRC cells, we also detected a time-dependent reduction in c-MYC protein amounts after MEK inhibition (Figure 1b, Appendix A). Moreover, the combined inhibition of p38α and MEK reduced the c-MYC protein levels to a greater extent compared to every single treatment alone; this effect was already detectable after 24 h (Figure 1c). In order to exclude the potential off-target effects of p38α inhibition by SB202190, we ablated p38α expression with a specific siRNA in the HCT116 cells. Our results were in agreement with the data obtained with SB202190. Indeed, *p38α* genetic ablation resulted in a 50% reduction of c-MYC protein levels. Moreover, c-MYC protein almost disappeared upon the co-silencing of *p38α* and *MEK* with specific siRNAs for 48 h (Figure 1d, left panel). The silencing efficiency of these siRNAs is also shown in Figure 1d (right panel).

### 3.2. Characterization of the Mechanism of Action of p38α on c-MYC 

Since protein half-life is an important factor in protein homeostasis, and c-MYC mRNA and protein are both inherently unstable, with a half-life of about 30 min [31,32], we investigated c-MYC regulation in a short-term experiment. We treated HT29 cells with SB202190 and/or PD98059 for 30 min and found that at this short time point, p38α and ERK do not regulate c-MYC transcription but affect its protein levels. Indeed, we could not detect any changes in c-MYC mRNA expression after either single or combined inhibition (Figure 2a, left panel); instead, we observed a significant decrease in the c-MYC protein levels after inhibition of the p38α or the MEK pathway, and an even greater effect after a combined inhibition of both cascades (Figure 2a, right panel).

Having control of protein abundance involves the regulation of both macromolecular synthesis (transcription and translation) and degradation (RNA decay and proteolysis) [33,34,35]. An analysis of the proteins at a single time point provides a snapshot of steady-state protein levels without revealing the relative contributions of synthesis and degradation. However, the contribution of each step from synthesis to degradation can be inferred by comparing the abundance before and after inhibiting specific components of the process [36]. In order to investigate whether p38α and ERK affect c-MYC translation, we used cycloheximide (CHX) at a final concentration of 0.2 mM and collected cells after 0 to 4 h. Cell lysates were then subjected to immunoblotting with an anti-c-MYC antibody and normalized to β-actin, a commonly used control in CHX chase assays. This experiment showed that p38α and ERK do not affect the c-MYC protein translational process because decreases in protein abundance following the addition of CHX can be confidently attributed to protein degradation (Figure 2b). 

As mentioned above, c-MYC stability is regulated at the post-translational level by phosphorylation at two conserved residues, S62 and T58 [12,37,38]. The ratio of T58 and S62 phosphorylation controls the c-MYC ubiquitination and turnover in cells. In response to the growth signals, ERK phosphorylates c-MYC on S62, increasing its stability and oncogenic activity. When the growth signals cease, S62-phosphorylated c-MYC can interact with GSK3β, which phosphorylates c-MYC at T58 [12,37,39]. Doubly phosphorylated c-MYC is then recognized by the PIN1 isomerase, which catalyzes the isomerization of the S62-proline 63 bond from a cis conformation to a trans conformation [38]. This isomerization allows PP2A to de-phosphorylate c-MYC at S62. Phosphorylation at T58 but not at S62 makes c-MYC prone to subsequent recruitment of ubiquitin ligase E3 and promotes initiation of its proteasome-dependent degradation [12,37,38,39].

Based on these post-translational regulation mechanisms, we treated HT29 cells with the GSK3β inhibitor, TWS119, or the PIN1 inhibitor, PiB, in combination with SB202190. The results obtained from this analysis showed that the c-MYC degradation induced by p38α inhibition is independent of PP2A and GSK3β (Figure 2c and Appendix A). Previous work from our group showed that *p38α* genetic depletion, or the pharmacological blockade of its kinase activity, induces autophagy—a process that can be involved in protein degradation [30]. To investigate whether c-MYC degradation after p38α inhibition is regulated by autophagy-associated mechanisms, we evaluated the c-MYC levels following inhibition of the cytoplasmic degradation process by treating HT29 cells with 3-methyladenine and bafilomycin (two autophagy inhibitors), in addition to the p38α inhibitor SB202190. No significant changes were observed in the c-MYC levels compared to the inhibition of p38α alone, suggesting that c-MYC degradation after p38α inhibition occurs through a different mechanism (Figure 2c and Appendix A).

It is well known that the deregulation of the ubiquitin-proteasome pathway is critical for c-MYC protein accumulation in cancers where no c-MYC amplification is observed [40]. Hence, to elucidate the mechanism underlying c-MYC destabilization induced by p38α inhibition, we blocked its proteasome-dependent degradation using the proteasome inhibitor MG132. Interestingly, MG132 could rescue SB202190-mediated c-MYC protein destabilization (Figure 2d). Taken together, these data suggest that p38α may stabilize c-MYC protein by preventing its ubiquitination. 

### 3.3. Pharmacological Targeting of p38α and ERK as a Synthetic Lethality Approach

In order to assess the translational implications of our findings, we confirmed whether our data could be replicated with ralimetinib and trametinib, which are currently being investigated in several clinical trials for inflammatory diseases and cancer [27,41,42]. Importantly, various clinical trials are being conducted on CRC patients. 

For this purpose, we performed dose- and time-response experiments to analyze the effect of ralimetinib in CRC cells (HT29 and HCT116) compared to the SB202190 treatment. The HT29 and HCT116 cells were treated with increasing doses of ralimetinib (from 1 μM to 10 μM) [7]. The immunoblot analysis showed that 24 h treatment with 10 μM ralimetinib effectively inhibits p38α activity in both cell lines, as estimated by impairment of MK2 phosphorylation at threonine 334 (Figure 3a). We previously demonstrated that SB202190-mediated p38α inhibition reduces the viability of CRC cells by inducing cell death [21]. To assess whether ralimetinib promotes a comparable cell death response, the trypan blue staining scores were analyzed in the HT29 and HCT116 cells at various time points after treatment with SB202190 or increasing concentrations of ralimetinib. Our results confirmed that inhibition of p38α with increasing doses of ralimetinib induced a significant rise in cell death in both the HT29 and HCT116 cells. Furthermore, the cell death rate showed a nearly 20% increase after a 36 h treatment with 10 μM of ralimetinib compared to SB202190 (Figure 3b). Next, we performed cell viability and WST-1 assays in order to assess the cytotoxicity and cell growth. Our results showed that ralimetinib modulates cancer cell survival and proliferation in both CRC cell lines (Figure 3c,d, respectively). 

After testing ralimetinib’s effectiveness in our cellular model, we evaluated whether it affects c-MYC stability in the HT29 and HCT116 cells. Both cell lines showed a decrease in c-MYC protein levels after 24 h treatment with ralimetinib, similar to the reduction induced by SB202190 (Figure 4a).

Then, we used ralimetinib in association with PD98059. Our results revealed a decrease in c-MYC protein amounts upon a combined inhibition of p38α and MEK in the HT29 and HCT116 cells at different time points (Figure 4b). These results were further confirmed when the MEK/ERK pathway was inhibited by using trametinib, a biological drug currently used for targeted therapy [42] (Figure 4c and Appendix A). 

Next, we quantified the cell death responses in the HT29 and HCT116 cells treated for 48 h with ralimetinib and trametinib either alone or in combination. Analyses of trypan blue staining scores showed that the cell death rate increased nearly 40% upon treatment with ralimetinib, about 30% upon treatment with trametinib, and over 60% upon combined treatment (Figure 4d). Moreover, we performed WST-1 and cell viability assays in order to assess cell growth and cytotoxicity. Our results confirmed that ralimetinib and trametinib induce a decrease in cancer cell proliferation and survival (Figure 4e,f, respectively). We also investigated the biological impact of a co-treatment with ralimetinib and trametinib on CRC cell lines by analyzing Ki67 expression and annexin V staining by flow cytometry. Our results indicate that the combined inhibition of p38α and MEK promoted a greater decrease in Ki67-positive cells compared to each single treatment (Figure 4g and Appendix A), and this effect was associated with the induction of apoptosis, while no necrosis was observed (Figure 4h and Appendix A). In these experiments, treatments with SB202190 and/or PD98059, which we characterized in a previous report [21,22], were used as a control for the impact of p38α and/or MEK inhibition on CRC cells (Figure 4g,h and Appendix A). 

Furthermore, given the major role of c-MYC as a transcription factor in CRC initiation and progression [10], we evaluated the expression of c-MYC target genes by real-time PCR upon p38α and MEK inhibition in HT29 cells. c-MYC inhibits the expression of p21 and activates the transcription of Cyclin E, Cyclin A, and cdc25, which are all involved in cell-cycle progression [43,44,45]. Treatments with ralimetinib and/or trametinib for 24 h increased the expression of p21 and reduced the levels of Cyclin E, Cyclin A, and cdc25 (Figure 4i). In order to improve the translational impact of our results, we evaluated the potential of the combined treatment with ralimetinib and trametinib in a CRC-stem cell (CRC-SC) tumorsphere model. Two lines of patient-derived CRC-SCs, grown as tumorspheres, were treated with ralimetinib and/or trametinib and subjected to immunoblot analysis. Our results revealed a decrease in c-MYC protein levels upon inhibition of p38α or MEK. This effect was even greater upon the combined inhibition of both kinases. Moreover, we detected the activation of the apoptotic pathway in CRC-SC tumorspheres treated with ralimetinib and/or trametinib, as shown by immunoblotting for cleaved PARP (Figure 4j). Again, a combined treatment with both inhibitors exhibited a higher pro-apoptotic activity compared to each single treatment alone (Figure 4j).

### 3.4. p38α Interacts with and Phosphorylates c-MYC

In order to better characterize the mechanism of action of p38α on c-MYC, we performed an in vitro binding assay between full-length GST-tagged p38α recombinant protein and full-length c-MYC recombinant protein, using HIS-tagged β-catenin as a positive control [22]. Our results showed that p38α directly interacts with c-MYC in vitro (Figure 5a). Then, we evaluated whether endogenous p38α interacts with c-MYC. Immunoprecipitation of whole-cell lysates with an antiserum against p38α or c-MYC, followed by immunoblotting, revealed that p38α is a molecular partner of c-MYC in HT29 CRC cells (Figure 5b). 

Since c-MYC protein stability is regulated by well-known phosphorylation signals in the N-terminal region [11,46,47], we searched for c-MYC residues that could be directly phosphorylated by p38α. To this end, we performed an in vitro kinase assay using purified proteins. Our results showed that active p38α could efficiently phosphorylate c-MYC, while inhibition of p38α activity by different concentrations of ralimetinib (0.01, 0.1, and 1 µM) significantly reduced c-MYC phosphorylation in vitro (Figure 5c).

Since 85% of the p38α-mediated phosphorylation sites described so far are Ser-Pro or Thr-Pro motifs [48], we performed an in silico phosphorylation prediction analysis using iPTMnet (https://research.bioinformatics.udel.edu/iptmnet; accessed on 14 April 2022), NETPHOS 3.1 (https://services.healthtech.dtu.dk/service.php?NetPhos-3.1; threshold ≥ 3, accessed on 14 April 2022), Phosphosite Plus (https://www.phosphosite.org/homeAction, only HTP data, accessed on 14 April 2022), and KynasePhos 2.0 (http://kinasephos2.mbc.nctu.edu.tw/, accessed on 13 December 2020). We identified two putative MYC phosphosites (S64 and S67) that were recognized by all four prediction servers (Figure 5d). To confirm the hypothesis that S64 and S67 are targeted by p38α, we performed a mass spectrometry (MS) analysis of c-MYC protein after phosphorylation by p38α in vitro. The E_54_LLPTPPLSPSRRSGLCSPSYVAVTPFSLRG_84_ peptide, obtained by proteolytic digestion with endoproteinase Asp-N, showed a mass increase of 80 Da, which corresponds to the weight of an additional phosphate at S64 or S67 (Figure 5e). 

### 3.5. Combined Inhibition of p38a and MEK1 Affects c-MYC Protein Stability in Preclinical Mouse Models

To evaluate the effects of the pharmacological blockade of both p38α and MEK/ERK on the c-MYC protein levels in vivo, we used the azoxymethane (AOM)/dextran sodium sulfate (DSS) colitis-associated-carcinoma murine model, which is considered a highly reproducible and reliable CRC model [49]. To this end, the animals were injected with the carcinogen AOM and then fed with DSS to induce colitis. Ten days after the last round of DSS, the animals were treated with daily intraperitoneal injections of SB202190 and oral PD0325901 (an MEK1/2 inhibitor) or with the vehicle alone (DMSO). After 10 days of treatment, the animals were sacrificed, and the explanted tissues were analyzed (Figure 6a). An immunohistochemical analysis of the colon sections showed high c-MYC nuclear positivity in vehicle-treated animals, whereas a decreased c-MYC expression was detected in the epithelial cells of animals subjected to the combined treatment with SB202190 and PD0325901 (Figure 6b,c). These results were further confirmed in tumor xenograft mice, a different in vivo murine model. Xenograft tumors were established by injecting HT29 cells into athymic nude mice. As soon as the tumors reached a measurable size, mice were divided into four groups, which were treated with the vehicle (DMSO), SB202190 alone, PD0325901 alone, or both SB202190 and PD0325901. Drug treatments were administered daily intraperitoneally (for SB202190) or orally (for PD0325901) for 10 days, and tumor volume and body weight were recorded every 2–3 days (Figure 6d). At the end of the treatments, the xenograft tumors were explanted, weighed, and subjected to immunohistochemical analysis for c-MYC expression. A strong reduction in tumor volumes was observed in the HT29-xenografted nude mice concomitantly treated for 10 days with SB202190 and PD0325901, compared to the tumors treated with the vehicle alone (DMSO) or the single agents (Figure 6e). As shown in Figure 6f,g, inhibition of p38α and MEK significantly reduced c-MYC nuclear positivity compared with the vehicle-treated tumors. Moreover, the colon sections derived from mice subjected to the combined treatment with SB202190 and PD0325901 showed no c-MYC nuclear expression (Figure 6f,g). These data validate, in vivo, the combined inhibition of p38α and MEK/ERK as a promising approach to reducing c-MYC-mediated CRC proliferation, metastatic dissemination, and chemoresistance.

### 3.6. Meta-Analysis on a Cohort of Colorectal Tumor Tissues Retrieved from TCGA PanCancer Atlas

In order to identify which patient subsets may most likely benefit from p38α pharmacological inhibition by ralimetinib, we performed a meta-analysis on a cohort of colorectal tumor tissues retrieved from The Cancer Genome Atlas (TCGA) PanCancer Atlas. This dataset encompasses the clinical data of 581 CRC patients. We stratified these 581 CRC patients based on the p38α protein expression z-score (RPPA data) and c-MYC protein expression z-score (RPPA data) and selected the subsets of patients with high p38α and high c-MYC expression (77 patients), or high p38α and low c-MYC expression (154 patients) to investigate the association between the expression levels of these two proteins and the patients’ prognoses. We found that high expressions of c-MYC protein in the subset of patients overexpressing p38α protein were associated with worse progression-free survival (PFS, *p*-value = 0.0157) and disease-free survival (DFS, *p*-value = 0.0301) (Figure 7a,b, respectively). These data suggest that p38α and c-MYC protein expression may be used as predictive biomarkers to select a specific subpopulation of CRC patients that could benefit from p38α pharmacological inhibition by ralimetinib.

## 4. Discussion

The current treatment for CRC is based on combination therapies, which in most cases include surgery, local radiotherapy, and chemotherapy [50]. Therefore, a promising approach involves finding critical molecular targets that are positively selected and essential for tumor growth. This concept is the basis of current research, and several compounds that target crucial signaling pathways are now under clinical trials. In CRC, c-MYC has been found to be overexpressed in up to 80% of sporadic cases—a result of its transcriptional activation mediated by β-catenin following APC inactivation and its post-translational stabilization mediated by ERK [10]. The MEK/ERK signaling pathway is involved in tumor initiation and progression by promoting cell proliferation and survival [51] and is being widely studied as a promising pharmacological target [52]. Intriguingly, MEK/ERK inhibition induces the phospho-activation of p38α, a kinase involved in CRC progression and cell migration [21]. The existence of a p38α/ERK crosstalk has been previously established since the inhibition of p38α also triggers ERK phospho-activation in different CRC cell lines [21]. 

Here, we found that the inhibition of p38α or MEK significantly reduces the c-MYC protein levels in CRC cell lines, regardless of their genetic background, and combined inhibition of both pathways has an even greater effect. Given the crucial role of c-MYC in intestinal tumorigenesis and its high turnover, we investigated the role of p38α and MEK in c-MYC regulation in a short-term experiment. Our results showed that a 30 min inhibition of p38α and MEK only affected c-MYC protein levels, while its mRNA expression remained unchanged. Since c-MYC has a significant impact on cell fate, it is not surprising that cancer cells have evolved sophisticated methods for ensuring that it maintains proper expression levels [12]. c-MYC protein expression is regulated both at the post-transcriptional (translation) and post-translational (protein stability) levels. We, thus, performed experiments to assess p38α’s and ERK’s contribution to these processes. Since we found that p38α and ERK are not involved in c-MYC translation, we assessed whether they affect c-MYC stability. Indeed, increased c-MYC stability in malignant tissues may explain the high levels of c-MYC protein observed in human tumors in the absence of c-MYC gene amplification or deregulated mRNA expression [53]. Our results showed that p38α and ERK actually influence c-MYC protein stabilization, but they do not do so by affecting the GSK3β or PP2A pathways or by regulating the autophagic response. Instead, we found that blocking proteasomal degradation rescues the c-MYC protein levels after p38α inhibition. These data revealed that p38α stabilizes c-MYC protein by preventing its ubiquitination. Moreover, we confirmed that p38α inhibition affects the c-MYC protein levels in vivo in two different murine models (AOM-DSS and xenograft mice). 

The potential efficacy of p38α and MEK/ERK combined inhibition—as a novel pharmacological strategy—is supported by the results obtained from CRC cell lines. These results highlight the importance of identifying protein kinases involved in tumorigenesis and tumor progression to shift the focus of cancer therapy to those drugs rationally designed to target tumor-specific pathways. Several clinical studies are now underway to collect information on the polypharmacological profile of kinase inhibitors. In light of this approach, we switched to the compounds that are currently in clinical trials for CRC, i.e., the p38α inhibitor, ralimetinib, and the MEK inhibitor, trametinib. After positively testing their inhibitory activity in our CRC cell lines, we confirmed that these pharmacological inhibitors are effective in increasing cell death response, decreasing the proliferation rate, and reducing c-MYC protein levels. 

As a transcription factor, c-MYC regulates the expression of 15% of all genes by binding to an enhancer box (E-box) sequence. Indeed, c-MYC coordinates a complex cellular response by which cells are set to progress through the cell cycle and proliferate [54]. Based on this assumption, we decided to ascertain the therapeutic potential of p38α and MEK inhibition by assessing the expression levels of c-MYC target genes. c-MYC regulates the transcription of several genes implicated in cell-cycle progression; specifically, it promotes the expression of Cyclin E, Cyclin A, and cdc25 and represses p21 [44]. Consistently, the decrease in the c-MYC levels, promoted by p38α and/or MEK inhibition, induced the downregulation of Cyclin E, Cyclin A, and cdc25 and the upregulation of p21 [44]. 

Despite the wide number of proteins that interact with c-MYC, only a few kinases have been shown to regulate its stability. These include ERK and CDK2, which phosphorylate c-MYC at S62, leading to its stabilization [38]. Instead, phosphorylation of c-MYC at T58 by GSK3β results in its degradation through the recruitment of the ubiquitin ligase Fbw7 and subsequent ubiquitin-mediated proteolysis [55]. Here, we found that p38α binds to c-MYC (in vitro and in cellulo) and phosphorylates it, and we identified S64 and S67 as putative phosphorylation loci in the c-MYC N-terminal domain. Intriguingly, S64 and S67 are located very close to the ERK target residue S62, suggesting that they may also be involved in protecting c-MYC from proteasomal degradation. Consistently, we demonstrated that p38α pharmacological inhibition by ralimetinib abrogates c-MYC phosphorylation. 

Despite the search for those molecularly targeted drugs capable of inhibiting the activity of protein kinases involved in cancer-specific features, a lack of efficacy in tumor progression or patient survival has been observed in several clinical trials using target-oriented anti-cancer compounds [56,57]. For example, a recent study evaluating the addition of ralimetinib to gemcitabine and carboplatin chemotherapy regimens in patients with recurrent platinum-sensitive ovarian cancer revealed only a modest improvement in PFS in these patients [58]. Indeed, it is becoming increasingly clear that targeted therapies are effective only in selected categories of patients whose tumors are characterized by distinctive molecular profiles. These setbacks mostly reflect the need for better knowledge of validated tumor biomarkers in order to both define specific therapeutic targets and properly select responsive patient populations. Although some relevant therapeutic responses have been associated with specific genomic markers, in the majority of cases, the assignment of patients to specific targeted treatments remains largely empiric or correlative at best. At present, no patients are “cured” of their disease by targeted therapies, despite the risk of treatment-related adverse events and an overall increase in medical costs. In order to exploit the specificity of new targeted compounds, it is necessary to tailor treatments to each patient’s cellular circuitry and design clinical protocols based on the molecular signature of each specific tumor. In this study, we sought to evaluate the use of p38α and c-MYC protein expression as biomarkers to select a specific subpopulation of CRC patients that can achieve adequate clinical benefit through p38α pharmacological inhibition by ralimetinib. Interestingly, our meta-analysis on a cohort of CRC tumors, retrieved from the TCGA PanCancer Atlas dataset, correlated p38α and c-MYC protein expression to prognosis, with high expressions of both p38α and c-MYC protein (~33% of CRC patients) being associated with worse DFS and PFS. Based on these findings, p38α and c-MYC may be used as markers of resistance and predictors of therapy response in CRC. Overall, our study has some limitations. Evidence of direct phosphorylation of c-MYC by p38α has only been obtained in vitro (kinase assay), the same being true for the serine residues putatively involved in the process (mass spectrometry analysis); thus, further studies will be necessary to better characterize these post-translational modifications and their consequences in vivo. 

## 5. Conclusions

Our previous and current data indicate that p38α contributes to c-MYC upregulation by promoting its transcription as a β-catenin chromatin-associated kinase and preventing the proteasome degradation of its protein product (Figure 8) [22]. Overall, our results provide insights into the dynamic processes that control c-MYC levels and support the potential of perturbing these mechanisms in cancer. Indeed, our findings suggest that the pharmacological inhibition of p38α and ERK, which support both c-MYC expression and protein stability, may prove as an effective therapeutic strategy for targeting CRC.

## Figures and Tables

**Figure 1 cancers-14-04840-f001:**
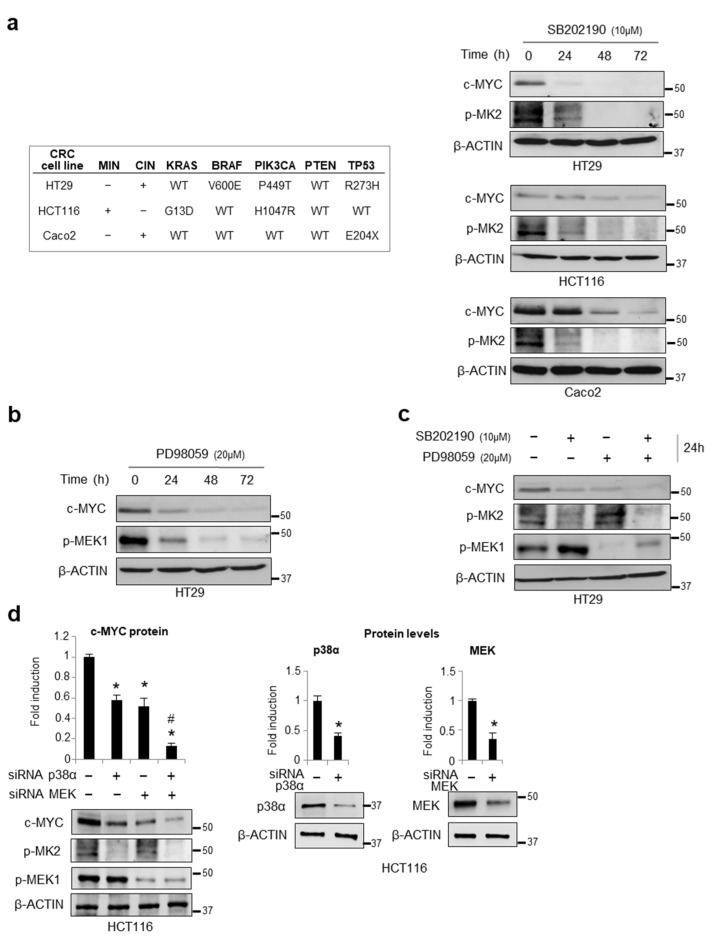
Effects of p38α/ERK crosstalk on c-MYC expression in CRC cells. (**a**) Left panel: Classification of the CRC cell lines used in this study by their microsatellite instability (MIN) and chromosome instability (CIN) phenotypes and the mutational status of critical cancer genes. Right panel: Immunoblotting showing c-MYC protein levels in HT29, HCT116, and Caco2 cells treated with SB202190 (10 μM) for up to 72 h. (**b**) Immunoblotting showing c-MYC protein levels in HT29 cells treated with PD98059 (20 μM) for up to 72 h. (**c**) Immunoblotting showing c-MYC protein levels in HT29 cells treated with SB202190 (10 μM) and/or PD98059 (20 μM) for 24 h. (**d**) Left panel: Densitometric analysis of c-MYC protein levels detected by immunoblotting in HCT116 cells transfected with p38α-specific and/or MEK-specific siRNAs. Right panel: Densitometric analysis of p38α and MEK protein levels detected by immunoblotting to show silencing efficiency in HCT116 cells transfected with p38α-specific or MEK-specific siRNAs, respectively. (**a**–**d**) β-actin was used for normalization; MEK1 activation (p-MEK1) and p38 activity (p-MK2) were analyzed to check treatment efficacy. (**d**) Statistical analysis was performed using Student’s *t*-test; * *p* < 0.05 vs. cells treated with control siRNAs, # *p* < 0.05: vs. single silenced cells. The presented results are representative of at least three independent experiments. Detailed information about Western Blot can be found at Appendix A.

**Figure 2 cancers-14-04840-f002:**
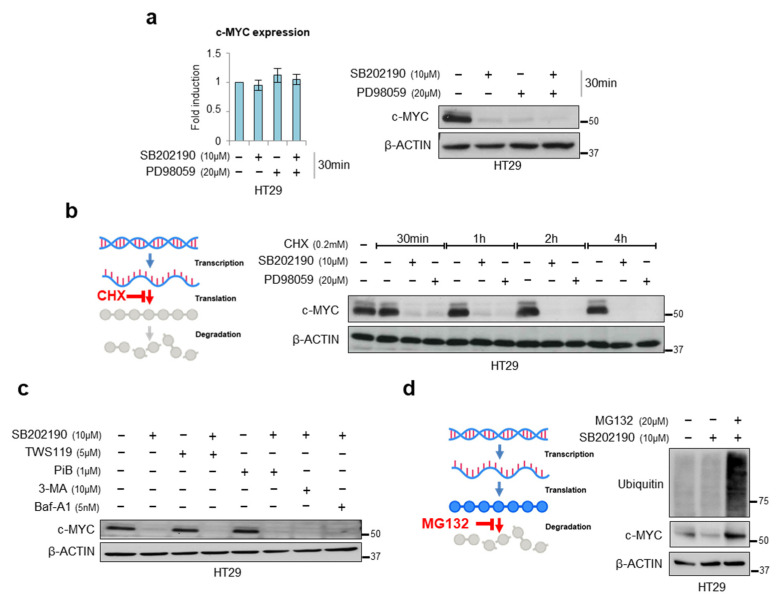
p38α stabilizes c-MYC protein. (**a**) Left panel: RT-PCR analysis of c-MYC mRNA levels in HT29 cells treated with SB202190 (10 μM) and/or PD98059 (20 μM) for 30 min. Right panel: Immunoblotting showing c-MYC protein levels in HT29 cells treated with SB202190 (10 μM) and/or PD98059 (20 μM) for 30 min. (**b**) Immunoblotting showing c-MYC protein levels in HT29 cells treated with the protein biosynthesis inhibitor cycloheximide (0.2 mM), whose action is schematically represented on the left, and SB202190 (10 μM) or PD98059 (20 μM) for 0 to 4 h. (**c**) Immunoblotting showing c-MYC protein levels in HT29 cells treated for 8 h with different inhibitors, i.e., the GSK3β inhibitor TWS119 (5 μM), the PIN1 inhibitor PiB (1 μM), the autophagic inhibitors 3-methyladenine (10 μM) and bafilomycin (5 nM), in combination or not with SB202190 (10 μM). (**d**) Immunoblotting showing ubiquitin and c-MYC protein levels in HT29 cells treated for 5 h with SB202190 (10 μM), with or without the proteasome inhibitor MG132 (20 μM), whose action is schematically represented on the left, for 1 h. CHX: cycloheximide, 3MA: 3-methyladenine, Baf-A1: bafilomycin. β-actin was used as a loading control. The presented results are representative of at least three independent experiments. Detailed information about Western Blot can be found at Appendix A.

**Figure 3 cancers-14-04840-f003:**
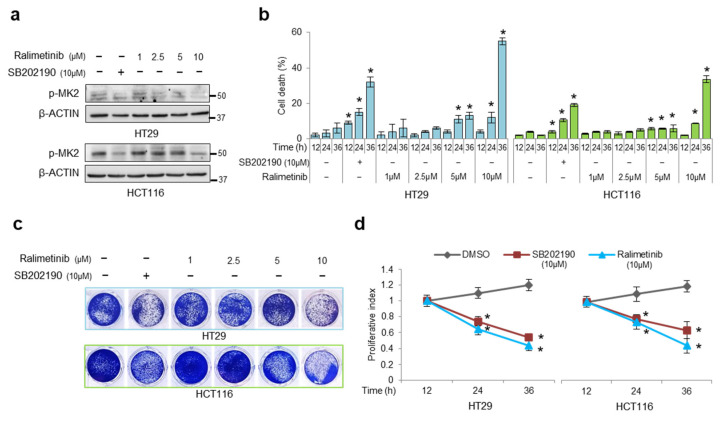
Treatment of CRC cells with ralimetinib or SB202190 decreases viability while increasing cell death. (**a**) Immunoblotting showing phosphorylated MK2 protein levels in HT29 and HCT116 cells treated with SB202190 (10 μM) or ralimetinib (1–10 μM) for 24 h. β-actin was used as a loading control. (**b**) Quantification of cell death by trypan blue staining in HT29 and HCT116 cells treated with SB202190 (10 μM) or ralimetinib (1–10 μM) for up to 36 h. (**c**) Cell viability assay on HT29 and HCT116 cells treated with ralimetinib (1–10 μM) or SB202190 (10 μM) for 36 h. (d) Proliferative index of HT29 and HCT116 cells treated with ralimetinib (10 μM) or SB202190 (10 μM) for up to 36 h, as determined by WST-1 assay. (**b**,**d**) Statistical analysis was performed using Student’s *t*-test; * *p* < 0.05 vs. untreated cells (**b**) or DMSO-treated cells (**d**). DMSO: dimethyl sulfoxide. The presented results are representative of at least three independent experiments. Detailed information about Western Blot can be found at Appendix A.

**Figure 4 cancers-14-04840-f004:**
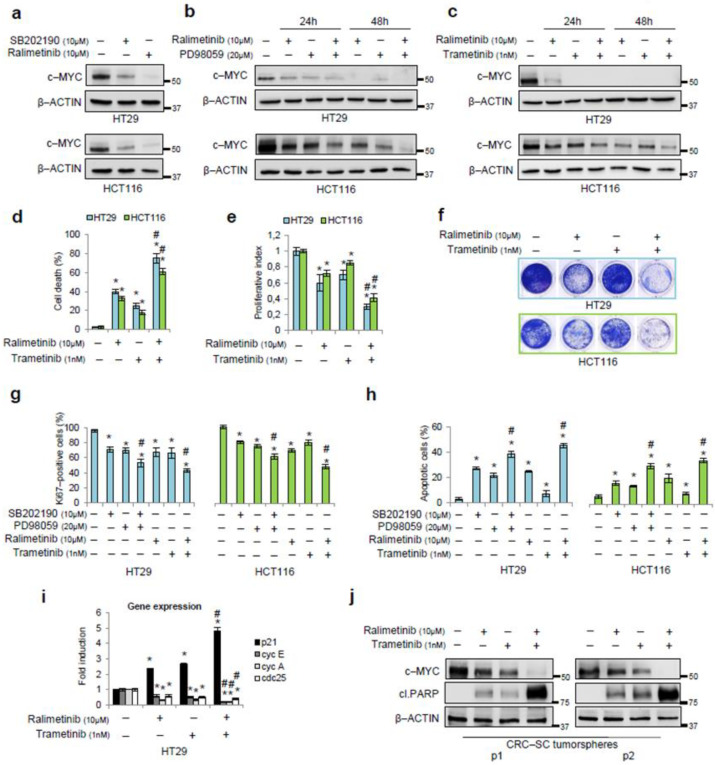
Pharmacological inhibition of p38α and MEK as a synthetic lethality approach. (**a**) Immunoblotting showing c-MYC protein levels in HT29 and HCT116 cells treated for 24 h with SB202190 (10 μM) or ralimetinib (10 μM). (**b**) Immunoblotting showing c-MYC protein levels in HT29 and HCT116 cells treated for up to 48 h with ralimetinib (10 μM) and/or PD98059 (20 μM). (**c**) Immunoblotting showing c-MYC protein levels in HT29 and HCT116 cells treated for up to 48 h with ralimetinib (10 μM) and/or trametinib (1 nM). (**d**) Quantification of cell death by trypan blue staining in HT29 and HCT116 cells treated with ralimetinib (10 μM) and/or trametinib (1 nM) for 48 h. (**e**) Proliferative index of HT29 and HCT116 cells treated with ralimetinib (10 μM) and/or trametinib (1 nM) for 48 h, as determined by WST-1 assay. (**f**) Cell viability assay on HT29 and HCT116 cells treated with ralimetinib (10 μM) and/or trametinib (1 nM) for 48 h. (**g**) Graphs summarizing the percentage of Ki67-positive cells, as determined by flow cytometry analysis, in HT29 and HCT116 cells treated with SB202190 (10 μM) and/or PD98059 (20 μM) or with ralimetinib (10 μM) and/or trametinib (1 nM) for 48 h. (**h**) Graphs summarizing the percentage of apoptotic cells (early + late), as determined by flow cytometry analysis of annexin V staining, in HT29 and HCT116 cells treated as in g. (**i**) RT-PCR analysis of the mRNA levels of the c-MYC target genes p21, Cyclin E, Cyclin A, and cdc25 in HT29 cells treated with ralimetinib (10 μM) and/or trametinib (1 nM) for 24 h. (**j**) Immunoblotting showing c-MYC and cleaved PARP protein levels in two lines of patient-derived CRC-SCs grown as tumorspheres (#9 and #40) treated with ralimetinib (10 μM) and/or trametinib (1 nM) for 48 h. (**a**–**c**,**j**) β-actin was used for normalization. (**d**,**e**,**g**–**i**) Statistical analysis was performed using Student’s *t*-test; * *p* < 0.05: vs. untreated cells, # *p* < 0.05: vs. corresponding single treatment; cl.PARP: cleaved PARP; p1: patient 1-derived CRC-SC; p2: patient 2-derived CRC-SC. The presented results are representative of at least three independent experiments. Detailed information about Western Blot can be found at Appendix A.

**Figure 5 cancers-14-04840-f005:**
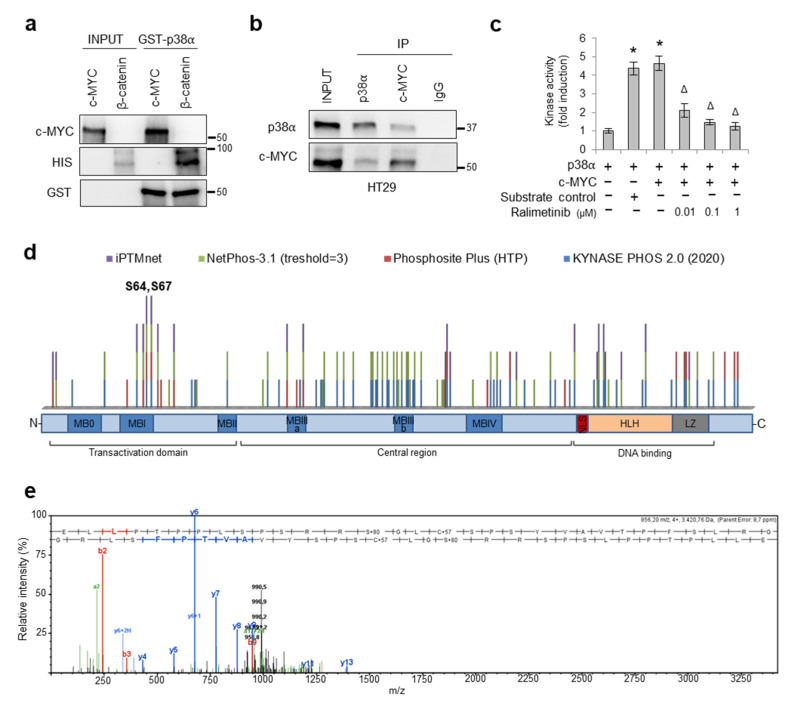
p38α interacts with and phosphorylates c-MYC. (**a**) In vitro binding assay between GST-p38α fusion protein and c-MYC recombinant protein. Bound proteins were analyzed by immunoblotting using anti-GST, anti-HIS, and anti-c-MYC antibodies. HIS-β-catenin was used as a positive control. (**b**) Co-immunoprecipitation of endogenous p38α and c-MYC in HT29 cells. (**c**) In vitro kinase assay showing c-MYC phosphorylation by p38α in the absence or presence of ralimetinib at the indicated concentrations. Statistical analysis was performed using Student’s *t*-test, * *p* < 0.05 vs. active p38α; ^∆^
*p* < 0.05 vs. active p38α + c-MYC. (**d**) In silico phosphorylation prediction meta-analysis. Four in silico prediction servers were used to identify MYC consensus phosphorylation sites: iPTMnet (https://research.bioinformatics.udel.edu/iptmnet; accessed on 14 April 2022), NETPHOS 3.1 (https://services.healthtech.dtu.dk/service.php?NetPhos-3.1; threshold ≥ 3, accessed on 14 April 2022), Phosphosite Plus (https://www.phosphosite.org/homeAction, only HTP data, accessed on 14 April 2022), and KynasePhos 2.0 (http://kinasephos2.mbc.nctu.edu.tw/, accessed on 13 December 2020). (**e**) MS/MS spectrum of the double-charged precursor ion of E54LLPTPPLSPSRRSGLCSPSYVAVTPFSLRG84. MB: MYC Boxed conserved domain. NLS: nuclear localization sequence. HLH: helix–loop–helix domain. LZ: leucine zip domain. The presented results are representative of at least three independent experiments. Detailed information about Western Blot can be found at Appendix A.

**Figure 6 cancers-14-04840-f006:**
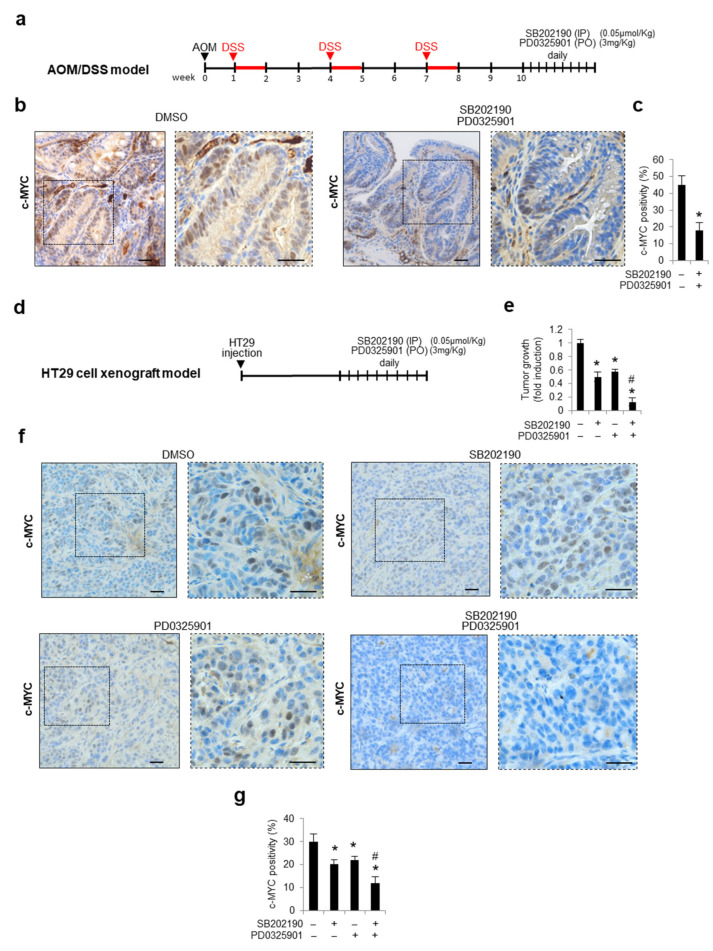
Combined inhibition of p38a and MEK affects c-MYC protein stability in preclinical mouse models. (**a**) Treatment scheme of AOM/DSS mice. (**b**,**c**) Immunohistochemistry analysis (**b**) and quantification (**c**) of c-MYC protein expression in colon tissue sections from AOM/DSS mice treated with DMSO or with SB202190 and PD0325901. (**d**) Treatment scheme of HT29-cell-xenograft mice. (**e**) Quantification of the volume of tumors explanted from HT29-xenografted mice treated with DMSO or with SB202190 and/or PD0325901. (**f**) Immunohistochemistry analysis of c-MYC protein expression in colon tissue sections from HT29-xenografted mice treated as above. (**g**) Quantification of c-MYC protein expression in colon tissue sections from HT29-xenografted mice treated as above. Magnification: 20×–40×. DMSO: dimethyl sulfoxide. Scale bar: 50 μm. IP: intraperitoneal injection. PO: oral administration. Statistical analysis was performed using Student’s *t*-test; * *p* < 0.05: vs. untreated cells, # *p* < 0.05: vs. corresponding single treatment. The presented results are representative of at least three independent experiments.

**Figure 7 cancers-14-04840-f007:**
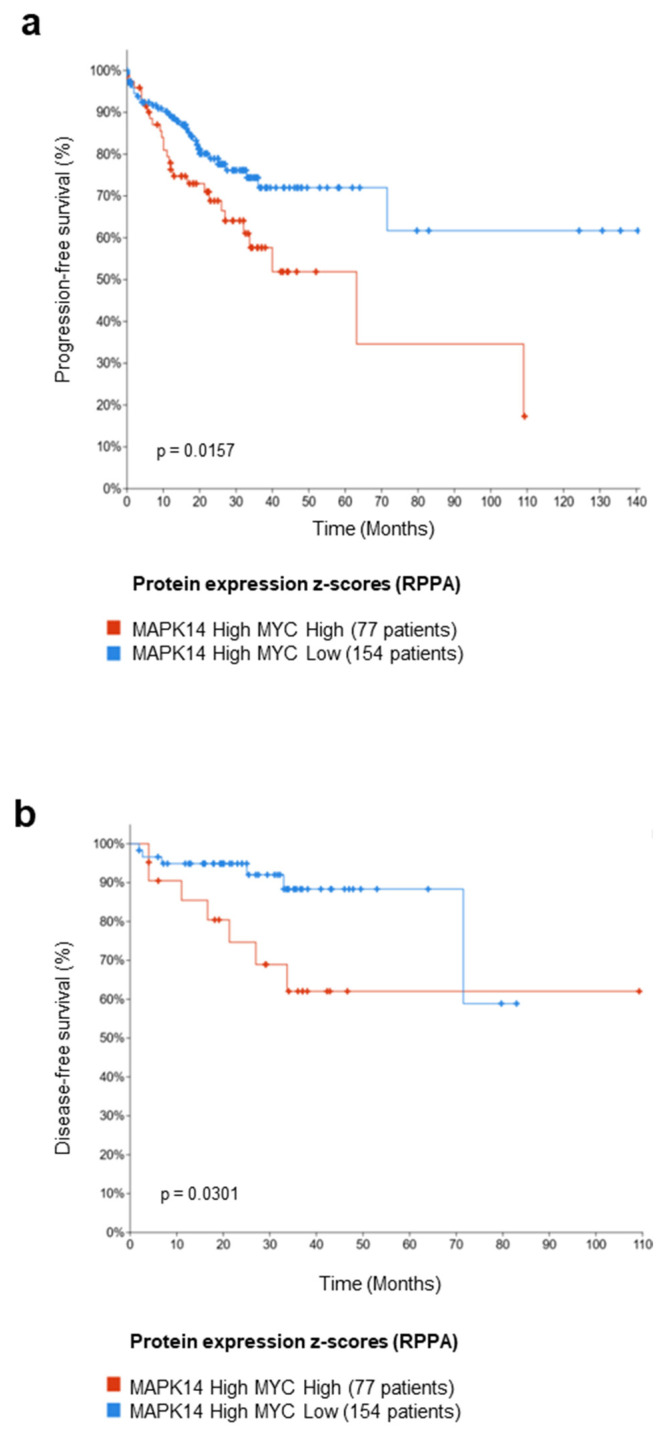
High protein expression levels of p38α and c-MYC are potential predictive biomarkers for therapy efficacy in CRC patients. (**a**,**b**) Kaplan–Meier curve of progression-free survival (PFS) (**a**) and disease-free survival (DFS) (**b**) in CRC patients as a function of p38α (MAPK14) and c-MYC (MYC) protein levels based on clinical data of 581 CRC patients retrieved from TCGA PanCancer Atlas.

**Figure 8 cancers-14-04840-f008:**
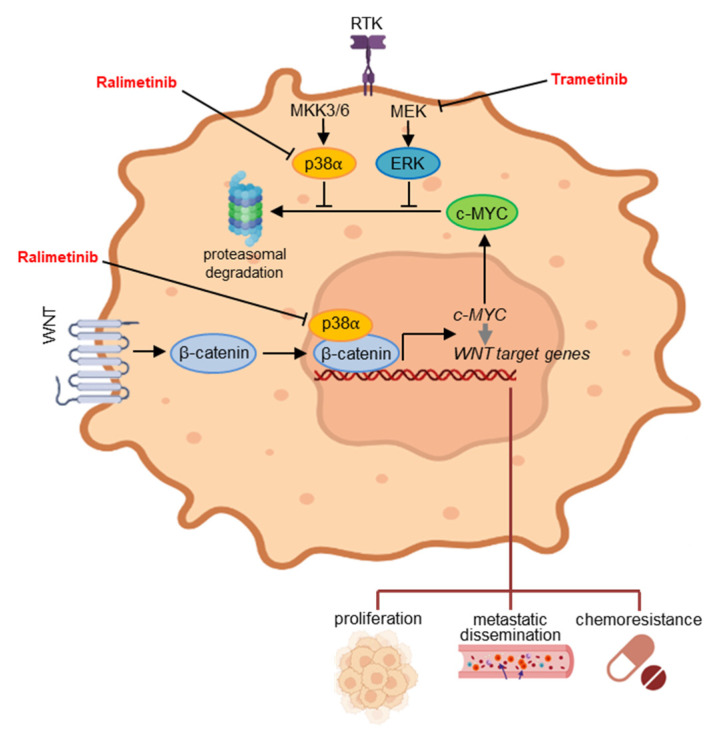
c-MYC is one of the most important factors in CRC and is maintained upregulated through β-catenin-mediated transcriptional activation and ERK-mediated post-translational stabilization. Our previous and current findings indicate that p38α, a kinase involved in CRC metabolism and survival, contributes to both mechanisms. As a result, the p38α inhibitor ralimetinib may prove effective in impairing cellular proliferation, reducing metastatic dissemination and overcoming chemoresistance in CRC.

## Data Availability

The colorectal adenocarcinoma dataset (coadread_tcga_pan_can_atlas_2018) analyzed for this study can be found in the TCGA—PanCancer Atlas (https://www.cbioportal.org, accessed on 2 May 2022).

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
