# Peer review of "c-MYC Protein Stability Is Sustained by MAPKs in Colorectal Cancer"

_cancers, 2022, doi:10.3390/cancers14194840_

Round 1
Reviewer 1 Report
In the work entitled “c-MYC Protein Stability Is Sustained by MAPKs in Colorectal Cancer,” the authors nicely describe the role of MAPKs (in particular p38a) in preserving colon cancer cell proliferation through the direct phosphorylation, and the consequent stabilization, of cMyc. The experiments performed support a central role of this kinase in maintaining Myc stability and provide important in vitro and in vivo evidence suggesting that its inhibition can be exploited as a novel pharmacological approach for those tumors expressing high levels of booth Myc and p38.
It is the opinion of the present reviewer that the data presented are of high quality, relevant for a broad readership and worth to be considered for designing novel therapeutic approaches for colorectal cancer; however, some minor and major concerns remain that should be addressed before publication.
Major revisions:
· In Fig.2C the authors should provide appropriate controls of the activity of the different drugs they used (i.e. PiB, TWS119, 3MA, Baf-A1)
· The entire work has been conducted on established cancer cell lines even if the results appear to be reproducible in all the cell lines tested, it would be nice to provide evidence of the role of MAPKs in a more clinically relevant setting such as patient-derived spheroids or organoids.
Minor revisions:
· At line 275 a reference is missing related to the statement “The existence of a p38α/ERK crosstalk and the identification of ERK-mediated c-MYC 274 protein stabilization mechanisms”
· From line 321 to line 326 and from line 360 to line 363, the detailed description of the structure and/or mechanisms of action of cycloheximide and of the MG132 can be removed. They are well known and widely used molecules.
· From line 330 to line 332. The statement is not clear and should be rephrased, the results presented in fig.2b nicely show that inhibition of p38 or MEK reduce Myc increase myc turnover.
· The author stated that they used the “PD988059” the correct name of the drug I supposed they used is “PD98059”. They also interchangeably refer to this drug as MEK or ERK inhibitor (see line 277, 313, 441) while it is a specific MEK-1 inhibitor that does not directly inhibit ERK1 or ERK2.
Author Response
Comments and Suggestions for Authors (Reviewer 1)
In the work entitled “c-MYC Protein Stability Is Sustained by MAPKs in Colorectal Cancer,” the authors nicely describe the role of MAPKs (in particular p38a) in preserving colon cancer cell proliferation through the direct phosphorylation, and the consequent stabilization, of cMyc. The experiments performed support a central role of this kinase in maintaining Myc stability and provide important in vitro and in vivo evidence suggesting that its inhibition can be exploited as a novel pharmacological approach for those tumors expressing high levels of booth Myc and p38.
It is the opinion of the present reviewer that the data presented are of high quality, relevant for a broad readership and worth to be considered for designing novel therapeutic approaches for colorectal cancer; however, some minor and major concerns remain that should be addressed before publication.
We are grateful to the Reviewer for this positive general comment. We addressed his/her concerns in the point-by-point response below.
Major revisions:
- In Fig.2C the authors should provide appropriate controls of the activity of the different drugs they used (i.e. PiB, TWS119, 3MA, Baf-A1)
We thank the Reviewer for this observation. In this amended version of the manuscript, we included appropriate controls to confirm the activity of the different drugs used in Fig. 2C. The results of these immunoblots are now depicted in Supplementary Figure 2.
- The entire work has been conducted on established cancer cell lines even if the results appear to be reproducible in all the cell lines tested, it would be nice to provide evidence of the role of MAPKs in a more clinically relevant setting such as patient-derived spheroids or organoids.
We are grateful to the Reviewer for this suggestion, as it allowed us to improve the translational impact of this work. We performed the suggested experiment, whose results are now depicted in new Figure 4j.
Minor revisions:
- At line 275 a reference is missing related to the statement “The existence of a p38α/ERK crosstalk and the identification of ERK-mediated c-MYC 274 protein stabilization mechanisms”
We thank the Reviewer for this observation. We added two references: PMID: 22579651; PMID: 11018017.
- From line 321 to line 326 and from line 360 to line 363, the detailed description of the structure and/or mechanisms of action of cycloheximide and of the MG132 can be removed. They are well known and widely used molecules.
We thank the Reviewer for this suggestion. We deleted from the manuscript the detailed description of the structure and/or mechanism of action of cycloheximide and MG132.
From line 330 to line 332. The statement is not clear and should be rephrased, the results presented in fig.2b nicely show that inhibition of p38 or MEK reduce Myc increase myc turnover.
We thank the Reviewer for this suggestion. We rephrased the sentence in order to clarify this point.
The author stated that they used the “PD988059” the correct name of the drug I supposed they used is “PD98059”. They also interchangeably refer to this drug as MEK or ERK inhibitor (see line 277, 313, 441) while it is a specific MEK-1 inhibitor that does not directly inhibit ERK1 or ERK2
We are grateful to the Reviewer for this observation. We corrected the typo in the name of the drug and the inaccuracy in its specificity throughout the manuscript and the figures.

Reviewer 2 Report
Your manuscript entitled 'c-MYC Protein Stability Is Sustained by MAPKs in Colorectal Cancer' is highly relevant and scientifically sound which can attract great interest from the scientific community in the field.
However, there is much room for improvement in the quality and content of the manuscript. Please consider the following points, to begin with.
Methods
2.4. Proliferation assays: further details required
2.5. Immunoblot analysis: further details required
2.8. Immunohistochemistry: further details required
2.10. In vivo studies: 10 million cells/mouse is too high. Here in our lab we regularly use 2.5million with a 100% engraftment rate. And in many other labs use 5 million. The medium you used for the inoculation of the cancer cell was not stated. How you have measured and calculated tumor volume is also not clear. As for other methods, this section also requires a clear explanation of what you have done so that others can repeat your work without any difficulty.
2.11. Colony formation assay: There have not been 48 h colony formation assays in the literature. What you have measured here is just a kind of relative viability/proliferation of cells in the presence of different treatments. For colony formation, one must follow the cell growth for at least 5-6 cycles of doubling which may last 7-14 days in most cases based on the proliferation rate of the cell line under study.
2.16. Dataset analysis: better to include correlation between expression of p38/ERK and c-MYC
2.17. Quantification and statistical analysis: what is this t-tail test mean? do you think this is enough to compare all the data sets you have got in the manuscript? Sometimes you may need to compare two groups' mean, other times more than that. So how could the student t-test alone provide you with the best statistical comparison for your data?
3. Results
Figure1: concentration of compounds should be indicated in a standard unit. Primary target inhibition readout must be included in addition to change in the level of c-MYC which is an effect of certain target downregulation such as p38 or ERK for all western blot data (e.g. as in Figure 1D).
The same must be applied throughout the manuscript to make it easily understandable to the readers
Figure 4g. These data need to be prepared for publication grade. The graph on the left side is not readable even at 200x magnification; the gating is not very clear. Repeat the experiment, present the figure as supplementary data and only show the bar graphs here to save space.
Figure 4f. There is clearly a compensation issue here. You need to repeat this experiment and do appropriate compensation. Otherwise, the data presented herein didn't reflect the true effect of your treatments even if there is desired effect. In the figure legend, you also stated apoptosis = early + late + dead. This is not correct mathematics!
Figure 6: is DMSO your vehicle in the actual experiment?
All the immunohistochemistry data need to be quantified(enumerated) and presented using a bar graph with an error bar included and compared using the appropriate statistical method.
In addition, tumour growth curve must be included for Figure 1b to show the efficacy difference among the groups and determine if the observed efficacy can track well with the change in the protein level of c-MYC in the immunohistochemistry analysis.
In section 3.6, I think the manuscript would be further benefited if data showing the correlation of p38 and c-MYC expression is mined and presented using a graph rather than the common survival analysis which wasn't even the main goal of the present work.
Limitations of this work must be explicitly stated under the discussion section.
In your conclusion, you have stated that p38α contributes to c-MYC upregulation both by promoting its transcription as a β-catenin chromatin-associated kinase and by preventing proteasome degradation of its protein production. However, on lines 592 and 593 you have discussed that your results showed that 30-minute inhibition of p38α and ERK only affected c-MYC protein levels, not its mRNA expression. How can one reconcile these two contrasting arguments?
Author Response
Comments and Suggestions for Authors (Reviewer 2)
Your manuscript entitled 'c-MYC Protein Stability Is Sustained by MAPKs in Colorectal Cancer' is highly relevant and scientifically sound which can attract great interest from the scientific community in the field.
We are grateful to the Reviewer for this positive general comment. We addressed his/her concerns in the point-by-point response below.
However, there is much room for improvement in the quality and content of the manuscript. Please consider the following points, to begin with.
Methods
2.4. Proliferation assays: further details required
2.5. Immunoblot analysis: further details required
2.8. Immunohistochemistry: further details required
We thank the Reviewer for this observation. As suggested, we added further details in the relevant Materials and Methods sections.
2.10. In vivo studies: 10 million cells/mouse is too high. Here in our lab we regularly use 2.5million with a 100% engraftment rate. And in many other labs use 5 million. The medium you used for the inoculation of the cancer cell was not stated. How you have measured and calculated tumor volume is also not clear. As for other methods, this section also requires a clear explanation of what you have done so that others can repeat your work without any difficulty.
We thank the reviewer for this comment. Although we are aware that other labs use fewer cells for xenograft experiments, in our previous studies (Chiacchiera et al. Cell Death Differ.2009; Chiacchiera et al. Cancer Lett 2012; Grossi et al. Cancer Biol Ther. 2012; Germani et al. Cancer Lett 2014) we regularly used 10 million cells/mouse without any problem. As suggested by the Reviewer, we added further details in the relevant Materials and Methods section.
2.11. Colony formation assay: There have not been 48 h colony formation assays in the literature. What you have measured here is just a kind of relative viability/proliferation of cells in the presence of different treatments. For colony formation, one must follow the cell growth for at least 5-6 cycles of doubling which may last 7-14 days in most cases based on the proliferation rate of the cell line under study.
We thank the Reviewer for this observation and we agree that “colony formation assay” is not appropriate for the described experiment. We thus renamed this experiment as “cell viability assay” throughout the whole manuscript.
2.16. Dataset analysis: better to include correlation between expression of p38/ERK and c-MYC
We thank the Reviewer for raising this important point. In the previously submitted version of the manuscript, we assessed p38α and c-MYC protein expression in colorectal cancer patients’ tumors for the purpose of therapeutic targeting with p38α pharmacological inhibition by ralimetinib. As reported in the manuscript, a higher expression of both p38α and c-MYC protein was associated with worse disease-free survival (DFS) and worse progression-free survival (PFS), suggesting that p38α and c-MYC may be used as markers of resistance and predictors of therapy response in CRC patients. Assessing the correlation between p38α/ERK and c-MYC expression is an interesting suggestion, but the results can be confounded by the complex mechanisms regulating c-MYC proto-oncogene deregulation in CRC, which include transcriptional activation by β-catenin following APC inactivation, post-traslational stabilization and activation by cyclin-dependent kinases (CDKs) mediated-phosphorylation, post-translational stabilization by ERK-mediated phosphorylation, and post-translational stabilization by p38α-mediated phosphorylation. Anyway, as suggested by the Reviewer, we evaluated the correlation between p38α and c-MYC expression by using proteomics data from the Clinical Proteomic Tumor Analysis Consortium (CPTAC) of 110 colon cancer patients (Vasaikar S, et al. Cell. 2019 May 2;177(4):1035-1049.e19. PMID: 31031003). Specifically, processed proteomics data were downloaded from the LinkedOmics portal (Vasaikar SV, et al. Nucleic Acids Res. 2018 Jan 4;46(D1):D956-D963. PMID: 29136207; http://linkedomics.org/cptac-colon/). In order to evaluate the role of p38α and ERK in c-MYC post-translational stabilization – and hence its upregulation -, we examined the correlation of p38α phosphosite (T180; Y182) and ERK1 phosphosite (T202; Y204) abundance with c-MYC phosphosite (T58; S62) abundance. For this analysis, the average values of p38α (T180; Y182), ERK1 (T202; Y204), and c-MYC (T58; S62) phosphorylation sites were used to quantify phosphoprotein abundance in 59 patient tumors for which values of c-MYC phosphorylation sites were available for analysis. The correlation between the average levels of p38α-ERK and c-MYC phosphorylation was determined using Pearson’s correlation coefficient. This analysis did not reveal a positive correlation between p38α phosphorylation sites (T180; Y182) and c-MYC phosphorylation sites (T58; S62) (Pearson’s r = 0.11; p = 0.42) (Figure 1a Rebuttal) nor between ERK1 phosphorylation sites (T202; Y204) and c-MYC phosphorylation sites (T58; S62) (Pearson’s r = 0.094; p = 0.48) (Figure 1b Rebuttal). This proteomic correlation analysis between p38α, ERK, and c-MYC confirms that c-MYC stabilization - and hence its upregulation - depends on multiple mechanisms contributing to clonal diversity and intratumor heterogeneity of CRC. It should be noted, however, that these results are based on the interrogation of a single phosphoproteomic dataset including a limited number of samples. Interrogation of further phosphoproteomic datasets of CRC patient cohorts will likely yield additional information regarding the correlation between p38α/ERK and c-MYC expression.
2.17. Quantification and statistical analysis: what is this t-tail test mean? do you think this is enough to compare all the data sets you have got in the manuscript? Sometimes you may need to compare two groups' mean, other times more than that. So how could the student t-test alone provide you with the best statistical comparison for your data?
We thank the Reviewer for this observation. In the previously submitted version of the manuscript, we erroneously indicated “Student’s t-tail test” (Figure 1, Figure 3, Figure 4, Figure 5, and paragraph 2.17 “Quantification and statistical analysis”) instead of “Student’s t-test”. Thus, in this amended version of the manuscript, we replaced “Student’s t-tail test” with “Student’s t-test”. This statistical test has been applied to compare the mean values obtained from two groups. Indeed, in our experiments, we did not carry out statistical analyses on multiple groups (e.g., analysis of variance (ANOVA)) since we did not compare the effect of cell treatments among several groups. Specifically, Student’s t-test has been used to evaluate the differences between observations in a treated cell line versus the same cell line in control conditions (i.e., untreated or vehicle-treated cells) (e.g. Figure 3b: the effect of cell death after 12 hours of SB2022190 and/or ralimetinib treatment in HT29 cells was compared with untreated HT29 cells maintained in culture for the same time).
- Results
Figure1: concentration of compounds should be indicated in a standard unit. Primary target inhibition readout must be included in addition to change in the level of c-MYC which is an effect of certain target downregulation such as p38 or ERK for all western blot data (e.g. as in Figure 1D).
The same must be applied throughout the manuscript to make it easily understandable to the readers
We thank the Reviewer for these comments. In this amended version of the manuscript, we added the concentration of the compounds in each figure. Moreover, we included the immunoblots showing the expression levels of c-MYC, p38α, and MEK proteins after treatment with siRNAs against p38α and MEK (new Figures 1d). The data from these immunoblots have been used to plot the graphs reported in Figure 1d.
Figure 4g. These data need to be prepared for publication grade. The graph on the left side is not readable even at 200x magnification; the gating is not very clear. Repeat the experiment, present the figure as supplementary data and only show the bar graphs here to save space.
We are sorry about the quality of the figures accompanying the previous version of our manuscript. In our previous submission, we separately uploaded PDF files with high-resolution figures in accordance with Cancers author guidelines, but something might have gone wrong. We now confirmed that high-resolution figures have been correctly uploaded with this revised version of the manuscript, and, as suggested by the Reviewer, we moved the plots data to new Supplementary Figure 3.
Figure 4f. There is clearly a compensation issue here. You need to repeat this experiment and do appropriate compensation. Otherwise, the data presented herein didn't reflect the true effect of your treatments even if there is desired effect. In the figure legend, you also stated apoptosis = early + late + dead. This is not correct mathematics!
We thank the Reviewer for these comments. We re-performed the suggested experiments. In this amended version of the manuscript, the results are depicted in new Figure 4h and Supplementary Figure 3b. We also corrected the error in the figure legend.
Figure 6: is DMSO your vehicle in the actual experiment?
DMSO is used as a vehicle in the experiment displayed in Figure 6. In this amended version of the manuscript, this is clearly specified in the relevant Materials and Methods and Results sections.
All the immunohistochemistry data need to be quantified(enumerated) and presented using a bar graph with an error bar included and compared using the appropriate statistical method.
We thank the Reviewer for this comment. As suggested, we quantified immunohistochemistry data and presented them as bar graphs in new Figure 6.
In addition, tumour growth curve must be included for Figure 1b to show the efficacy difference among the groups and determine if the observed efficacy can track well with the change in the protein level of c-MYC in the immunohistochemistry analysis.
We are grateful to the Reviewer for this suggestion. In this amended version of the manuscript, we included a comparison of tumor growth data among the various study treatments. The results are now depicted as a bar graph in new Figure 6.
In section 3.6, I think the manuscript would be further benefited if data showing the correlation of p38 and c-MYC expression is mined and presented using a graph rather than the common survival analysis which wasn't even the main goal of the present work.
Limitations of this work must be explicitly stated under the discussion section.
We thank the Reviewer for this recommendation. As suggested, we included the limitations of our work at the end of the Discussion section.
In your conclusion, you have stated that p38α contributes to c-MYC upregulation both by promoting its transcription as a β-catenin chromatin-associated kinase and by preventing proteasome degradation of its protein production. However, on lines 592 and 593 you have discussed that your results showed that 30-minute inhibition of p38α and ERK only affected c-MYC protein levels, not its mRNA expression. How can one reconcile these two contrasting arguments?
We are grateful to the Reviewer for this observation as it gives us the opportunity to clarify an important point of our work. We previously demonstrated (PMID: 33767160) that p38α acts as a β-catenin chromatin-associated kinase involved in the activation of β-catenin target gene transcription. Importantly, we showed that p38α pharmacological inhibition or its genetic ablation induce the downregulation of β-catenin target genes, including c-MYC. Here, we showed that inhibition of p38α for 30 minutes leads to destabilization of c-MYC protein but does not affect its mRNA expression. Consistently, we observed that a longer p38α inhibition is required to inhibit the mechanism of action associated with β-catenin transcription machinery. Specifically, we found that after three hours of p38α pharmacological inhibition, both c-MYC mRNA and protein levels were downregulated (data not shown).

Round 2
Reviewer 1 Report
The authors addressed all the concerns raised by the present reviewer. It is my opinion that the quality of the manuscript has improved significantly and can be considered for publication in cancers.
Author Response
Dear Dr. Sabrina Song,
we are pleased to submit the amended version of our work “c-MYC protein stability is sustained by MAPKs in colorectal cancer” (cancers-1837921) that we would like to have considered for publication in Cancers as part of the Special Issue "Innovations in Early Cancer Diagnostics and Therapeutics". We addressed below all the comments raised by the Reviewers, mainly by performing new experiments and by responding/clarifying or adding novel sentences in the text. Comments and Suggestions for Authors (Reviewer 1)
Reviewer 1
The authors addressed all the concerns raised by the present reviewer. It is my opinion that the quality of the manuscript has improved significantly and can be considered for publication in cancers.
We thank the reviewer for this positive comment. Comments and Suggestions for Authors (Reviewer 2)
Reviewer 2
Dear Authors,
I have received your revised version of the manuscript. You have very much improved the quality of your work and addressed the majority of the provided comments.
We thank the reviewer for this positive comment. We addressed his/her concerns in the point-by-point response below.
Please add western blot data for p38 and pMEK each time you present western blot data for c-MYC throughout your paper. It is not acceptable to present these data separately; please try to align these three markers in the same western blot data. In the first place, such data should be generated from the same experiment. One needs to know what should really happen to c-MYC if you down-regulate those proteins (p38 and MEK, say), genetically or using chemical inhibitors. For example, there should be western blot bands for p38, and MEK in each of Fig 1a, b, c, and d parallel to that of c-MYC.
We have performed the suggested western blotting panel and the results are now depicted in new Figure 1a, b, c,d.
In supplementary figures 3a and b, clearly, there is the issue of gating and compensation, respectively. In 3a, proliferation in trametinib is lower than that of trametinib+ralimetinib. But it was misrepresented as if the combination has a better anti-proliferative effect because of the gating. In my opinion, red vs blue doesn't necessarily discriminate a negative population from a positive population in your graph. Fix these issues and compare your data once again.
We have re-performed the suggested experiments in the previous first round comments. In this second round comments, the results are checked once again confirming the data obtained.

Reviewer 2 Report
Dear Authors,
I have received your revised version of the manuscript. You have very much improved the quality of your work and addressed the majority of the provided comments.
Please add western blot data for p38 and pMEK each time you present western blot data for c-MYC throughout your paper. It is not acceptable to present these data separately; please try to align these three markers in the same western blot data. In the first place, such data should be generated from the same experiment. One needs to know what should really happen to c-MYC if you down-regulate those proteins (p38 and MEK, say), genetically or using chemical inhibitors. For example, there should be western blot bands for p38, and MEK in each of Fig 1a, b, c, and d parallel to that of c-MYC.
In supplementary figures 3a and b, clearly, there is the issue of gating and compensation, respectively. In 3a, proliferation in trametinib is lower than that of trametinib+ralimetinib. But it was misrepresented as if the combination has a better anti-proliferative effect because of the gating. In my opinion, red vs blue doesn't necessarily discriminate a negative population from a positive population in your graph. Fix these issues and compare your data once again.
Author Response

(The authors gave the same response as above.)

Round 3
Reviewer 2 Report
Dear Authors,
I am satisfied with the current version of your manuscript.